# Why do you choose this program?—A decision-making model of medical students based on grounded theory

Yan Wang[1,2]*, Shutong He[3], Yanmin Chen[1]

1 Institute of Medical Education, Nanjing Medical University, Nanjing, Jiangsu, China, 2 School of Education, Nanjing University, Nanjing, Jiangsu, China, 3 Kangda College, Nanjing Medical University, Lianyungang, Jiangsu, China

* wangyannjmu@njmu.edu.cn

**Data Availability Statement:** All relevant data are within the paper and its Supporting Information files.

**Funding:** The author(s) received no specific funding for this work.

## Abstract

### Background

This study aims to investigate the reasons behind the decline in the number of applicants and dropouts from N University's reformed program, which includes increased research experience, an optimized curriculum, and other benefits. The ultimate goal is to identify areas for improvement and make the program more appealing to potential students.

### Methods

This study utilized the Grounded Theory approach, conducting semi-structured in-depth interviews and applying data collection, coding, and the constant comparative method. As a result, a decision-making model for college students was constructed.

### Results

Following the initial stages of individual expectation formation, which include inducement and self-efficacy, and the subsequent stage of value assessment, individuals reach a decision. Throughout this process, the individual's circumstances and surroundings continue to influence their decision-making. Additionally, the decision-making procedure follows a Hierarchy Pyramid of Educational Needs. Our findings show that job prospects and continuing education are the primary factors influencing interviewees' decisions. However, it is important to note that individuals may place varying levels of importance on these factors. Additionally, the preferences and priorities of teachers, such as their commitment to research guidance, curriculum development, and maintaining fairness in examinations, can also play a role in shaping these decisions.

### Conclusion

To attract more talented individuals to research-oriented programs, universities should provide more job and higher education opportunities, reform the curriculum thoroughly, and enhance teachers' teaching devotion.

**Competing interests:** The authors have declared that no competing interests exist.

# 1. Introduction

## 1.1 Background

Top-notch talent enrollment and cultivation have recently been a hot topic in China's higher education. To attract more students to careers in scientific research areas via undergraduate research [1], N University launched a major undergraduate program reform named the "National Key Laboratory" Program (hereinafter referred to as NKL) in 2018. NKL utilizes the National Key Reproductive Medicine Laboratory's resources to expose undergraduates to more scientific research as early as possible.

NKL initiated numerous reforms. Traditional 5-year basic medicine or preventive medicine program enrollment will be conducted during the nation's College Entrance Examination after high school graduation. The enrollment of the NKL, which started at the end of the first year in college, is not limited by one's original program. After the first year in school, students who meet the requirements of the university can choose to join the program. After passing the tests, one can officially become a member of NKL. Students in NKL can choose to quit at the beginning of the school year, and the ones who quit will automatically return to the traditional program of basic medicine or preventive medicine.

Traditional 5-year basic medicine or preventive medicine program's curriculum comprises 4-year theoretical learning and 1-year thesis writing. Different from the traditional one, NKL compresses the theoretical teaching of the traditional program and adds a scientific research training rotation plan in six of the national key laboratories in the school, which lasts for roughly 2 years. The change in the curriculum provides more opportunities for undergraduates to make sense of scientific research. In addition, the run-through program also provides every student with much higher scholarships and more postgraduate admission opportunities. As illustrated in Fig 1, after selecting students from traditional programs(left part), the blue part explains the reforms of NKL; the orange part represents other benefits of NKL compared with traditional programs; the light grey part represents the quit rule of NKL.

## 1.2 Raising questions

However, in the past three years, the number of applicants for the NKL has gradually declined, and some students even chose to quit. This phenomenon has aroused our interest: how do medical students make educational decisions? Basic scientific research is arduous, requiring

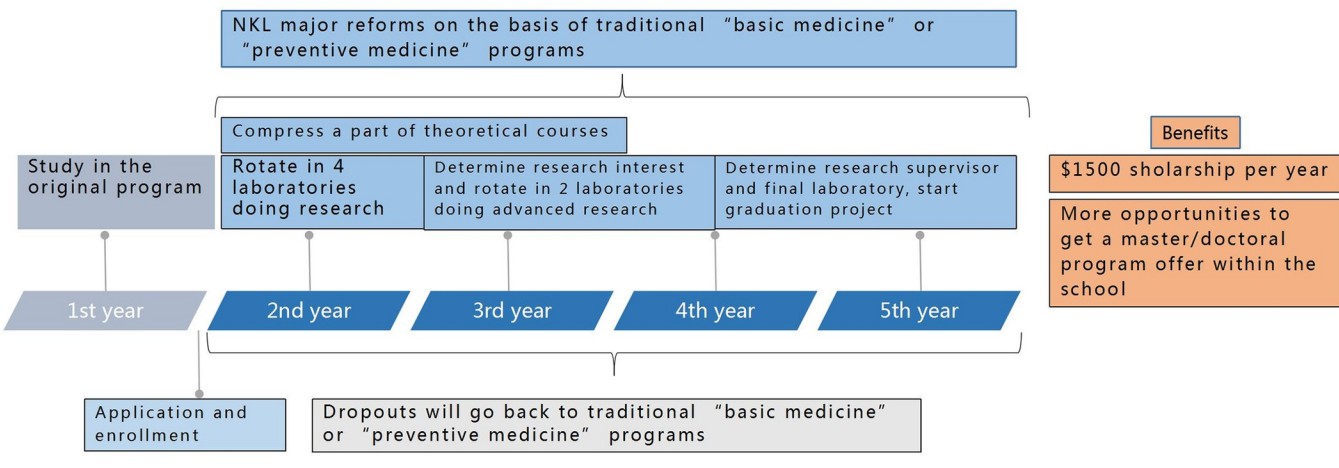

**Fig 1. NKL introduction.**

students to have a strong interest and willpower. It is particularly important to know how to make top-notch talent training programs more attractive to students.

### 1.3 Literature review

The Expectancy-Value Theory of Educational Psychology holds that people will anticipate the outcome and consider the value of an activity before deciding to engage in it. Pintrich [2] believes that the choice of tasks can reflect students' motivation to learn. Atkinson believes that people's behavioral tendencies are determined by motivation, probability of success, and inducement value. Eccles and Wigfield et al [3]. refined Atkinson's theory by arguing that value beliefs are related to choice behavior and that any factor affecting an individual's expectations and values can affect the choice, performance, and persistence of an individual's achievement behavior. Eccles [4] defines the achievement value of a task in terms of four components: achievement value, intrinsic value, utility value, and cost. Achievement value refers to a task confirming the stable or core components of an individual's self-schema. Intrinsic value refers to the enjoyment or subjective interest an individual obtains from an activity. Practical value refers to the association of the task with the individual's long-term goals. Cost refers to the negative effects of engaging in an activity. In this sense, it is a rational decision-making model from the perspective of motivation [5].

In the field of decision-making behavior model research in higher education, research mainly focuses on the construction of clinical specialty decision-making models in medical education, and other scholars have conducted preliminary research on the construction of decision-making behavior models. Jacqueline found that medical students are more willing to choose a rational decision-making model rather than an emotional decision-making model [6]. Joana has studied the factors that affect college students' decisions to drop out of school. In addition to the main influencing factor of academic performance, gender, course type, and whether or not to voluntarily apply for school all affect students' decisions to drop out [7]. Social Cognitive Career Theory proposes that individual career choices are influenced by self-efficacy and career outcome expectations [8]. Dawson holds that gender, clearance of objective, and major classification (mainly male or female) will affect the selection of major [9]. Johnson finds the effects of gender and self-efficacy on college students' program choices [10]. In addition, factors such as school, family, and society will affect the decision-making process of college students' choices [11]. Du Yang investigated the influence of major cognition and policy cognition on the program selection of undergraduates [12].

Therefore, it can be found that there is little qualitative research on students' decision-making models at present. Although the above-mentioned research has helped us find out many possible reasons behind different program choices, qualitative research is still needed to be done to see the bigger picture of a student's program selection process. The author will use the grounded theory method to try to explore the whole process of making decisions on what to learn.

## 2. Methods

### 2.1 Research methods

We used the grounded theory approach in qualitative research introduced by Stauss and Corbin [13]. Grounded theory is very suitable for the descriptive explanation of some social phenomena with the characteristics of "process" and "interaction" [14]. Its theoretical methods have been given more and more attention by scholars in the field of medical education. As a research methodology, grounded theory is a qualitative research method that uses a systematic procedure to develop the theory inductively for a certain phenomenon, rather than make

deductive assumptions in existing theories [15]. Because the research topic focused on the "process" of various major decisions of students learning in the NKL and we tried to find the "interactive" factors that affect the decision-making of college students, the research topic was very suitable for the grounded theory method. Therefore, the author planned to use the in-depth interview research method, data collection, coding, and constant comparative method [16] based on grounded theory to construct a model of college students' decision-making behavior illustrating the reasons for different choices.

Ethical approval for this study was obtained from the Board of Academic and Teaching Affairs of Nanjing Medical University. The work was carried out in accordance with the Declaration of Helsinki. Before the interview, informed consent was obtained from all subjects. During the interview, voluntary participants could skip the questions they did not want to answer. Participants could withdraw consent by informing researchers, and the data would be deleted.

## 2.2 Participant selection

The author selected research objects based on the principle of "purposive sampling" of grounded theory, that is, a case is selected because the case is very suitable for illustrating and expanding the interrelationship and logic between different constructs. Based on various decision-making opportunities in the procedure, we identified the interviewee standards as: 1. Students who met the NKL registration requirements (top 50% of grades, etc.) and were familiar with the NKL. 2. Students from NKL. 3. Dropouts from the NKL.

The researchers first issued a notice and then students would voluntarily sign up in 2021. The researchers then identified interviewees by double-checking students who met the standards and ensured all kinds of students were included. Authors had access to information that could choose suitable participants during data collection. All interviewees would be informed of "moral and ethical instructions" before the interview. They were aware that the face-to-face interview process would be recorded, written on the memos [17] and, kept confidential, and all materials would be used in an anonymous form, which reduced the possibility of them being too afraid to express true feelings. Interviewees might skip questions that made them feel uncomfortable and had the right to notify the researchers to delete all materials and related research whenever possible.

## 2.3 Data collection

The data collection for this study adopted the most common semi-structured interview method in grounded theory. First, an interview outline was formulated according to the educational psychology literature review and practical experience related to the decision-making behavior pattern of college students. The main interview questions were: 1. Was there any point in the learning process that encouraged or discouraged you from applying to NKL? 2. Tell me about the process of choosing the college major, and what do you think is the biggest difference between the NKL and the traditional program? 3. Tell me about your feelings since enrollment or anything that made you happy, dissatisfied, or impressed. 4. (For NKL students) Has the NKL met your expectations? The interviews were conducted by the authors. The average duration of the semi-structured interviews was 1.5 hours, and the author compiled a total of about 240,000 Chinese words of interview records.

This study followed the principle of "theoretical saturation." That is, the interviewees were selected for interviews, and the interviews were carried out sequentially. When the new cases could not provide more information or it was difficult for researchers to learn more information from the new cases, the case selection ended.

All three authors participated in the interviews and each interview was conducted and facilitated by two of the authors. Yan Wang (male) was the director of teaching evaluation centre

of Nanjing Medical University who had 8-year experience in higher education management and was an education doctor candidate at Nanjing University. Doctor Shutong He (male) was the provost of Kangda College who had over 15-year experience in higher education management and was a Ph.D. in higher education. Yanmin Chen (female) was a second-year student pursuing a master of public administration at Nanjing Medical University. During the data collection process, all three authors would continuously communicate with each other to control bias. Before the study, no relationship was established between participants and interviewers and the interviewers knew nothing but the major and gender of the participants before the interview.

All interviews were conducted in the classroom. No one would be present apart from the participant and interviewers. Transcripts were not returned to participants because interviewers could choose to listen to the recordings again to assure correctness.

## 2.4 Data analysis

Coding is an important method in grounded theory. It is a bridge between data and theory. This study employed a three-level coding process, namely, open coding, axial coding, and selective coding [18]. The first step was labeling the data, and then regrouping and categorizing important concepts with similar content and meaning. The second step was the interview summary sheet. After each case interview was over, the author's ideas were written down on the collated text materials to form an independent summary sheet. The last step was constant comparison analysis. By comparing the earlier and later interviews and comparing the data from this interview with other similar research data, we tried to anchor attributes and dimensions specific to each category. The method of constant comparison analysis was carried out throughout the course of this study. Yan Wang and Yanmin Chen coded the data.

**2.4.1 Open coding.** Open coding is to analyze the interview data and select the concepts that best represent the relevant connotations to categorize the interview data. After the preliminary analysis was over, all concepts were classified to dig out higher-level categories. At the same time, we carried out repeated comparisons and analyses. In the process of categorization, the author eliminated the initialization concepts with repetitions of less than 3 and those with little relevance to the research topic. Through open coding, this study formed 142 concepts and 40 initial categories.

**2.4.2 Axial coding.** Axial coding is to link the categories formed by open coding and combines related theories to analyze the potential logical relationship of related categories. First of all, the 40 categories formed in the open coding stage were further clustered and summarized to determine whether there were potential links in the meaning and logical relationship of each category. Second, we determined the subordinate relationship among categories and developed the main category and subcategory. Third, we established the logical structure among categories. Through analysis, further clustering and induction were carried out on the initially formed categories, and a total of 7 main categories and 18 subcategories were summarized.

**2.4.3 Selective coding.** Selective coding refers to the process of selecting core categories, connecting them with other categories, verifying these relationships, and finally completing the conceptualized categories. In this study, the program decision-making model of college students was divided into three key parts: "generating expectation," "value assessment," and "external influence".

## 3. Results

A total of 12 respondents were interviewed in this study, and the demographics of the interviewees are shown in Table 1. No one dropped out during the interviews.

**Table 1. Demographics of the interviewees.**

| Number | Gender | Grade | Program before NKL | Remark |
|--------|--------|-------|--------------------|--------|
| P1 | Male | 4 | Nursing | NKL student |
| P2 | Female | 4 | Basic Medicine | NKL student |
| P3 | Male | 4 | Preventive Medicine | NKL student |
| P4 | Male | 3 | Preventive Medicine | NKL student |
| P5 | Male | 4 | Preventive Medicine | NKL student |
| P6 | Female | 2 | Biomedical Engineering | NKL student |
| P7 | Male | 3 | Nursing (current program | A student who met the requirement but did not apply for NKL |
| P8 | Female | 4 | Basic Medicine | NKL dropout |
| P9 | Female | 2 | Management | NKL student |
| P10 | Female | 4 | Nursing | NKL dropout |
| P11 | Male | 4 | Basic Medicine | NKL dropout |
| P12 | Female | 1 | Nursing (current program) | A student who met the requirement and was ready to apply for NKL |

## 3.1 Category interpretation

Through selective coding, regarding the core category of "college students' educational decision-making behavior model", seven main categories are delineated which were derived from the data and checked by participants (Table 2).

**3.1.1 Inducement.** Inducement can be divided into primary motivation and external incentives. Primary motivation refers to something that an individual wants to actively promote from the very beginning. For example, P1 wanted to leave the nursing program upon admission because of barriers to choosing the nursing profession as a man, which is seen in many countries [19].

*I had a strong motivation to switch programs from the very beginning of grade one because I did not like nursing*[P1].

External incentives refer to the effects of things that happen unexpectedly.

*Originally, I didn't have an interest in basic scientific research. But after hearing a lecture about the deeds of female scientists in high school, I was inspired and decided to devote myself to basic scientific research*[P2].

**3.1.2 Self-efficacy.** Self-efficacy refers to a person's subjective judgment on whether one can successfully achieve something. Efficacy expectations refer to conjectures about the outcome of a corresponding decision and subjective judgment of one's ability to achieve the goal.

*Before deciding to sign up for the NKL, I will consult the teachers and senior students to judge whether it will succeed or not based on my situation. If the possibility of failure is greater, I will choose to give up*[P12].

*I don't care if I can successfully join NKL and graduate from it. However, I am concerned that I cannot keep up with the training progress so I will not be able to obtain scientific research thinking. This negatively affected my expectations and motivation to apply for the NKL*[P6].

**3.1.3 Extrinsic value.** In the category of extrinsic value, short-term target value refers to material value, academic performance, knowledge acquisition, scientific research skills, and other things that can significantly elevate an individual in the short term. The short-term target value was recognized by many interviewees.

**Table 2. Axial coding result.**

| Number | Main Category | Subcategory | Initial Category |
|---|---|---|---|
| 1 | Inducement | Inducement | Primary motivation |
| | | | External incentives |
| 2 | Self-efficacy | Efficacy expectations | Subjective judgments of self-ability and outcome |
| 3 | Extrinsic value | Short-term target value | Scholarship |
| | | | Competence improvement |
| | | Mid-term target value | Career planning |
| | | | Faster track |
| | | | A Higher education degree |
| | | Long-term target value | Scientific research ability and thinking |
| | | | The far-reaching impact of basic research on human beings |
| 4 | Intrinsic value | Interest | Explore interests and experience the unknown |
| | | Emotion value | Sense of pride, achievement, pleasure |
| 5 | Costs | Sunk costs | Costs of giving up the investment |
| | | Individual conflict | Economic cost |
| | | | Conflict with personality |
| | | | Study pressure |
| 6 | Individual situation | Proficiency | Knowledge |
| | | Economic | External financial assistance |
| | | Personality | Character, interests, and hobbies |
| 7 | Environment | Interpersonal environments | Learning (teachers, relatives, peers) |
| | | | Researching (teachers, peers) |
| | | | Living (relatives, peers) |
| | | Teaching environments | Management |
| | | | Curriculum, intensity, and teaching effectiveness |
| | | | Teacher |
| | | Social environments | Comments in society and on the Internet |

*The research rotation training at NKL has the potential to allow me to learn more research skills, which is very meaningful and attractive to me*[P4].

Mid-term goal value refers to things that individuals deem meaningful and valuable in the foreseeable future, such as career planning, higher degrees, and a faster training track. This part was deemed the most important part among all subcategories by many of the interviewees.

*The integration of undergraduate, master, and doctoral training at NKL can make it more possible for me to obtain a Ph.D. in a shorter time, which is of high value to me*[P3].

The long-term target value refers to the value of scientific thinking acquisition, the far-reaching influence of basic discipline research, and scientific research ability acquisition that cannot be obtained in a short time or that takes a long time to prove. Such things usually have a profound impact on an individual's life and even on human society, and this is what some consider valuable.

*Applied scientific research can only affect a part of society, while basic discipline scientific research can affect the underlying structure of the scientific research landscape, which is more meaningful. So I chose to apply for NKL*[P5].

**3.1.4 Intrinsic value.** Intrinsic value refers to individual interest and emotional value. Personal interest refers to an individual's interest in scientific research and a personality that likes to experience the unknown. Respondents with individual interests generally have higher primary motivations. Emotional value refers to the positive emotions such as the senses of pride, joy, achievement, and gain that are brought to an individual. The triggering of these emotions can make individuals attach a higher value to related things and decisions.

**3.1.5 Costs.** Costs refer to the possible negative impact of a decision that an individual encounters. Costs can be divided into sunk costs and conflicts with individuals. Sunk costs refer to the time and energy wasted if someone fails. It can also refer to the cost that an individual has invested before the decision, and giving up the decision will mean giving up the previous effort.

> *When I was thinking about quitting NKL, I was concerned about the time and energy invested upfront, which to some extent prevented me from deciding to quit*[P8].

Individual conflict refers to the clash between realistic decision-making and objective factors such as personal situation and environment. For instance, the conflict between long-term study without income and personal economic conditions, and the conflict between extroverted personality and "lonely" research, are both dissuasive. At the same time, it is also important to note that negative emotions and fewer career choices caused by NKL are also counted.

> *When I told my classmates that I was going to quit, they were not surprised at all because they all knew how depressed I was*[P10].

**3.1.6 Individual basis.** The individual basis continues to influence the whole process of individual decision-making. Individual basis refers to someone's profession, finance, personality, interest, etc. Proficiency refers to a collection of students' professional knowledge and ability, which has a great impact on intrinsic value. Finance refers to the economic conditions of an individual. Personality refers to personal character. Scientific research requires time-consuming, sitting-indoor experiments. This means that people with an outgoing personality will feel uncomfortable when doing basic scientific research, thus adding costs. Interest refers to a personal passion for research, which is highly related to intrinsic value.

**3.1.7 Environment.** The environment also continues to influence the whole process of individual decision-making. The environment can be divided into interpersonal environments and teaching environments. The interpersonal environment refers to the social environment that affects individual interpersonal relationships and has an unexpectedly important impact on decision-making. Something like the degree of approval from relatives and peers or family education will affect one's decision-making.

Teaching environment refers to the teacher, curriculum, teaching effect, and learning intensity, which will affect the individual's perception of value. In a learning environment, individuals often interact with and listen to teachers, relatives, and peers. In a research setting, individuals consult and listen to teachers and peers.

> *During my rotation, I met a postdoctoral researcher who became a friend of mine and told me a lot about scientific research……Our conversation strengthened my belief in what direction to take in the future*[P11].

Teachers' devotion affects the effectiveness of education greatly. NKL students have to rotate in different laboratories, but the evaluations of different rotation experiences are quite different. The authors find that teachers' guidance on experiments and devotion to the students greatly improved the students' evaluation and scientific research recognition level.

*Some well-funded laboratory teachers will teach students how to do experiments, and those kinds of laboratories are generally popular because word of mouth has already spread among students*[P10].

## 3.2 Model interpretation

In this study, the seven main categories are divided into three key parts: "generating expectation," "value assessment," and "external influence." The relationship framework is defined as the program decision-making model of medical students, as shown in Fig 2.

**3.2.1 Generating expectation.**   Students need an inducement to initiate the formation of individual expectations before making decisions. Many psychologists believe that both motivation and external stimuli evoke behavior. After the inducement, it enters the stage of self-efficacy and evaluates whether one is likely to succeed. Other similar studies have also confirmed that motivation and self-efficacy will affect the next step's willingness and behavior [20].

**3.2.2 Value assessment.**   In the value assessment stage, individuals assign and evaluate different values and costs respectively. The study found that the value assignment varies greatly among different types of people. Under normal circumstances, individuals who generate expectations with primary motivation focus more on intrinsic value, while those who generate expectations with external incentives tend to pay more attention to extrinsic values.

*My parents are both scientific researchers. Influenced by my family, I have been interested in scientific research since childhood. Completing tasks that contribute to society will satisfy me* [P2].

In P2's self-reported decision-making process, she made little assessment of the value of the short-term and medium-term goals but used a large amount of space to describe the intrinsic value that NKL brought. During her studies, the interviewee found that basic medical science

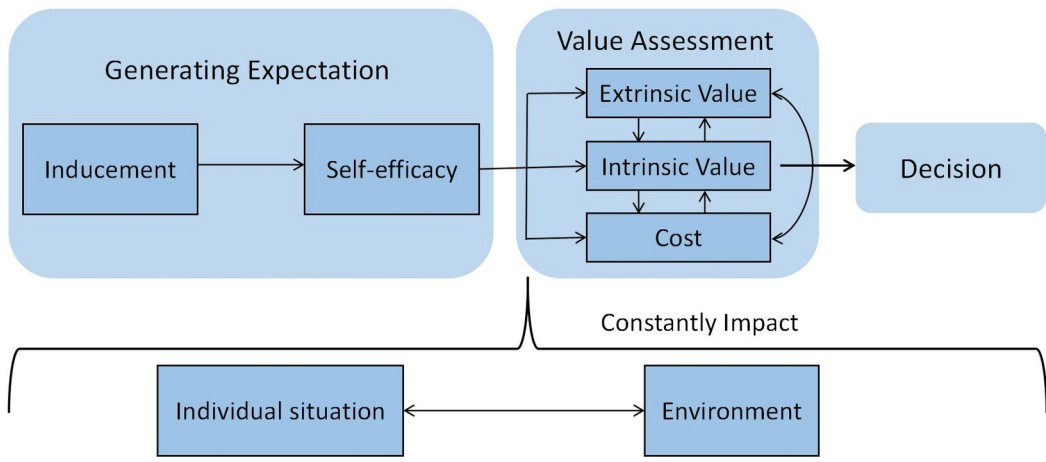

**Fig 2.  Program decision-making model of college students.**

research was not her interest, and finally decided to apply for postgraduate studies in the field of humanities and social sciences after graduation.

On the contrary, for respondents who are motivated by external incentives in their self-reported decision-making process, the proportion of intrinsic value was significantly lower than that of extrinsic value.

> *My priority is to find a job…….I didn't have a good idea of what scientific research was like at the time…….Because my family has only had one doctor so far, my father wanted me to continue studying, which prompted me to apply for NKL*[P1].

There also seems to be a slight difference between groups in their assessment of costs. Interviewees whose expectations were triggered by primary motivation focused on the conflict with individuals, while those who were influenced by external factors focused on sunk costs.

**3.2.3 Constant impact of outer factors.**   We find that an individual's situation can have an impact on the decision-making process. For instance, individuals with relatively weak economic backgrounds often have deep concerns about the long training systems of top-notch talents because it means that the student cannot provide financial support to the family for a long time. Therefore, the relatively short NKL is more attractive. Related research in China also finds a similar phenomenon [21].

> *Frankly speaking, the lengthy training system for doctors is not very friendly to students with weak economic foundations like me*[P5].

The environment also has a significant impact on individual decision-making processes. We find that, in addition to official information sources, the main source of information for most interviewees was the advice of teachers, relatives, and peers. Interaction between individuals and their environment is an important factor affecting individual behavior [22]. In the context of human interaction, respondents are constantly acquiring information and adjusting their own expectations and value judgments.

> *The experiences and feelings of senior students in NKL will affect my evaluation of it*[P12].

The teaching environment such as the devotion of teachers also affects the individual's decision-making process.

> *In the scientific research rotation, the devotion of different teachers varies greatly, and we all love to rotate in the laboratory, in which the supervisor guides students on research*[P8].

## 4. Discussion

### 4.1 Hierarchy pyramid of educational needs

The decision-making model of college students in this study has the obvious characteristics of the hierarchical structure of educational needs. In psychology, Maslow's Hierarchy of Needs theory is widely known, which describes human needs as a hierarchy within a process pyramid, with some needs taking precedence over others. In qualitative interviews, the authors find that students also have a hierarchy of educational needs when assessing values. When there is a conflict between low-level needs and high-level needs, students will give priority to meeting low-level needs. It will change the corresponding decision, the content of which is shown in Fig 3.

**Fig 3. Hierarchy pyramid of educational needs.**

We divide students' educational needs into four levels. The lowest level is graduation, which means getting a graduation certificate. The second level refers to the need to successfully find a job or further education upon graduation. The third level means that students acquire critical thinking, scientific research skills, and other things that are beneficial to the long-term development and growth of individuals. The fourth level is social development, which refers to the satisfaction of the core values in the student's self-schema and can be seen in P5.

*I hope that my research can promote the revitalization of national scientific research and even the progress and development of human society*[P5].

The author finds that the satisfaction of educational needs can greatly affect the individual's decision-making. P11 is an example.

*Although I am not very interested in basic scientific research, I still choose to stay in NKL because it has great advantages in getting a higher degree and facilitating personal development*[P11].

But later, P11 found out that the graduation research project should be completely in line with the reproductive medicine direction, otherwise, he would not be able to graduate. Therefore, when the needs of the two conflicted, P11 finally chose to drop out of NKL because he was afraid of not being able to graduate. Another roommate of P11 was also faced with a similar dilemma, but because he could still meet the graduation requirements successfully, after thinking for a long time, he finally decided to stay in the NKL. Individuals are very complex in the process of decision-making, and they will face various dilemmas. Similar dilemmas can also be seen in other interviewees. However, almost all decisions are in line with the order of the educational needs hierarchy pyramid during value assessment.

### 4.2 Main characteristics of students' decision-making pattern

First, we find that most interviewees place more emphasis on employment and further education. Studies in another discipline [23] and in medicine [24] proved the same point. The number of people who place more emphasis on their interest in scientific research at the undergraduate level is relatively low, and the economic advantages of this group are more significant. This is in accordance with the conclusion of relevant quantitative research in China [25]. In the sample of this study, more interviewees were motivated by external incentives than those motivated by primary motivation. Therefore, for top-notch students, the author assumes that the extrinsic value still has a greater impact than the intrinsic value. Thus, the unclear plan for further education opportunities at that time dissuaded many students from applying.

Second, different groups of people have different emphases. As mentioned above, individuals whose behavior is awakened by primary motivations are more sensitive to intrinsic value and individual conflict costs. On the contrary, individuals who are motivated by external incentives are more sensitive to extrinsic values and sunk costs.

> *I will follow whatever research direction that the tutor asks me to do, as long as I can successfully get the degree*[P4].

> *My supervisor wanted us to change the direction because our project direction was different from that of our supervisor…….. This conflict brought me a strong sense of anxiety and depression*[P10].

Third, different groups of people may have very different value assessments of the same thing. When conducting qualitative interviews with eligible students who were unwilling to apply for the NKL, the interviewees said that many people around them were reluctant to apply because they believed that basic research meant limiting their career possibilities. However, applicants for NKL generally believed that the acquisition of scientific research skills and thinking could enhance their competitiveness. Because these two groups of people have different evaluations of the same thing, they produce diametrically opposite decisions.

The fourth is the influence of family education. When conducting qualitative interviews, we find that family education plays an important role in the formation of critical thinking and scientific research thinking. However, at present, the author rarely reads papers elaborating on the influence of family education on students' thinking formation. At present, many studies still focus on classroom teaching. The cultivation of students' thinking by family members may become a better research direction in the future.

### 4.3 Limitations

The first limitation was the context. The research was conducted in the context of the education reform program NKL. Basically, participants were deciding whether to stay in the original program or transfer to the reformed NKL program. Therefore, the results might be influenced by this fact. The second limitation was participant selection. All participants came from the N University. This fact might impact the scope of the findings.

## 5. Conclusion

This research adopts the method of grounded theory to explore and establish a program decision-making model for college medical students. Under the joint action of many aspects, such as expectation generation, value evaluation, and the continuous influence of individuals and the environment in the model, students finally decide which program to apply to. Therefore,

to make the research-oriented program more attractive to top-notch talents, universities should provide more job and higher education opportunities, reform the curriculum thoroughly, and enhance teachers' teaching devotion.

## Supporting information

**S1 File. Raw data.**
(ZIP)

**S2 File. COREQ (COnsolidated criteria for REporting Qualitative research) checklist.**
(PDF)

## Author Contributions

**Conceptualization:** Yan Wang.

**Data curation:** Yan Wang, Yanmin Chen.

**Formal analysis:** Yan Wang, Shutong He, Yanmin Chen.

**Investigation:** Yan Wang, Yanmin Chen.

**Methodology:** Yan Wang.

**Project administration:** Yan Wang, Shutong He, Yanmin Chen.

**Resources:** Yan Wang.

**Supervision:** Shutong He.

**Validation:** Yan Wang, Shutong He.

**Visualization:** Yan Wang.

**Writing – original draft:** Yan Wang.

**Writing – review & editing:** Shutong He.

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
