## [Decision Letter · Decision Letter 0]

12 Jul 2023

PONE-D-23-12532Why do you choose this program?-a decision-making model of medical students based on grounded theoryPLOS ONE

Dear Dr. wang,

Thank you for submitting your manuscript to PLOS ONE. After careful consideration, we feel that it has merit but does not fully meet PLOS ONE’s publication criteria as it currently stands. Therefore, we invite you to submit a revised version of the manuscript that addresses the points raised during the review process.

We look forward to receiving your revised manuscript.

Kind regards,

Federica Canzan

Academic Editor

PLOS ONE

4. We note that Supplementary Figures 1, 2, 3, 4, 5, 6 and 7 in your submission contain copyrighted images. All PLOS content is published under the Creative Commons Attribution License (CC BY 4.0), which means that the manuscript, images, and Supporting Information files will be freely available online, and any third party is permitted to access, download, copy, distribute, and use these materials in any way, even commercially, with proper attribution. For more information, see our copyright guidelines: http://journals.plos.org/plosone/s/licenses-and-copyright.

1. You may seek permission from the original copyright holder of Supplementary Figures 1, 2, 3, 4, 5, 6 and 7 to publish the content specifically under the CC BY 4.0 license.

Reviewers' comments:

Reviewer's Responses to Questions

**Comments to the Author**

1. Is the manuscript technically sound, and do the data support the conclusions?

Reviewer #1: Yes

Reviewer #2: Yes

2. Has the statistical analysis been performed appropriately and rigorously? 

Reviewer #1: N/A

Reviewer #2: Yes

3. Have the authors made all data underlying the findings in their manuscript fully available?

Reviewer #1: Yes

Reviewer #2: Yes

4. Is the manuscript presented in an intelligible fashion and written in standard English?

Reviewer #1: Yes

Reviewer #2: Yes

5. Review Comments to the Author

Reviewer #1: Thank you for the opportunity to review your manuscript, it was a pleasure to read. I am thrilled you have taken the grounded theory method to apply to your research question. I appreciate the effort one makes when trying to adhere to all the required elements for qualitative research in general and grounded theory specifically. A few things to consider with revision: (a) the manuscript switches between future, past, and present tense which makes it confusing - for instance this study will and a paragraph later this study found. I would suggest switching to past tense for the methods section. (b) Under methods, you have only lightly cited your approach. While your first two citations assist the reader in knowing the type of research you are conducting, the rest of the methods are not defended in any way. You have also failed to disclose your philosophical stance and your researcher reflexivity or how you controlled your bias in this research. For citations, your three levels of coding, please cite all the literature that guided you in this approach. Also, I noticed two citations for your choice of methods Straub et al., 1992 and Glasser et al, 1968. Those two citations, while seminally defining the field, are 31 and 55 years old respectively. A lot has changed especially in the last decade. In fact, Glasser "remodeled" his GT approach 35 year later, in the early 2000s. I would suggest renewing your grounded theory and citing current literature (past 10-12 years). Some additional authors you might explore are Saldaña and/or Adu (coding), Charmaz, Bryant, Mills, Urquhart, De Chesnay (nursing), Glasser & Strauss (2017), and/or Flick. Finally, while realizing page limit restrictions, I would love to see more participant quotes that assisted you in reaching saturation. All that said, this seemed like a well conducted study that lead to important discussion and conclusions. Again it was an honor to review!

Reviewer #2: The authors of this study explores how medical students make educational decisions with a grounded theory approach. This research is well done, however, I have a few comments regarding reporting.

1. I recommend that the authors use the COREQ-checklist. This is a checklist of items that should be included in reports of qualitative research. Several are missing here.

2. When was the NKL launched?

3. Table 1 is a result and should be reported in the results section.

4. Some new results are reported in the discussion section.

5. A discussion about the limitations is missing.

6. PLOS authors have the option to publish the peer review history of their article (what does this mean?). If published, this will include your full peer review and any attached files.

Reviewer #1: **Yes: **June E Gothberg

Reviewer #2: **Yes: **Wim Peersman

---

## [Author Response · Author response to Decision Letter 0]

5 Aug 2023

1.We have formatted the manuscript according to the requirements listed in the <TITLE, AUTHOR, AFFILIATIONS FORMATTING GUIDELINES> and <MANUSCRIPT BODY FORMATTING GUIDELINES>.

2.We have deleted the ethics statement at the end of the manuscript and moved it to the Methods section.

3.We have included the caption for the supporting information at the end of the manuscript.

4.The supplementary Figures 1-7 were initially submitted to prove that the manuscript had been edited for language usage and had nothing to do with the content of the manuscript. Therefore, we chose to remove the figures from submission.

5.We reviewed the reference. It is complete and correct without citing any retracted article.

Reviewer #1:

1.We have switched to past tense for the Methods section.

2.First, we chose the way of team work to control bias in this research which has been added to the methods section. Second, we have updated the reference in the methods section according to the reviewer’s suggestion.

Reviewer #2:

1.According to the COREQ-checklist. We have added the items to the manuscript. The COREQ-Checklist has been added as supporting information.

2.NKL was launched in 2018. It has been added in the manuscript.

3.Table 1 has been moved to the result section.

4.We have moved some results from discussion to results and have added some content to the results section to avoid that some results are reported only in the discussion section.

5.We have added the limitation at the end of discussion section.

---

## [Decision Letter · Decision Letter 1]

4 Sep 2023

Why do you choose this program?-A decision-making model of medical students based on grounded theory

PONE-D-23-12532R1

Dear Dr. wang,

We’re pleased to inform you that your manuscript has been judged scientifically suitable for publication and will be formally accepted for publication once it meets all outstanding technical requirements.

Kind regards,

Federica Canzan

Academic Editor

PLOS ONE

Additional Editor Comments (optional):

Reviewers' comments:

Reviewer's Responses to Questions

**Comments to the Author**

1. If the authors have adequately addressed your comments raised in a previous round of review and you feel that this manuscript is now acceptable for publication, you may indicate that here to bypass the “Comments to the Author” section, enter your conflict of interest statement in the “Confidential to Editor” section, and submit your "Accept" recommendation.

Reviewer #2: All comments have been addressed

2. Is the manuscript technically sound, and do the data support the conclusions?

Reviewer #2: Yes

3. Has the statistical analysis been performed appropriately and rigorously? 

Reviewer #2: N/A

4. Have the authors made all data underlying the findings in their manuscript fully available?

Reviewer #2: Yes

5. Is the manuscript presented in an intelligible fashion and written in standard English?

Reviewer #2: Yes

6. Review Comments to the Author

Reviewer #2: The authors of this study explores how medical students make educational decisions with a grounded theory approach. The authors addressed my comments adequately.

7. PLOS authors have the option to publish the peer review history of their article (what does this mean?). If published, this will include your full peer review and any attached files.

Reviewer #2: **Yes: **Wim Peersman

---

## [Editor Report · Acceptance letter]

8 Sep 2023

PONE-D-23-12532R1 

Why do you choose this program?
-A decision-making model of medical students based on grounded theory 

Dear Dr. Wang:

I'm pleased to inform you that your manuscript has been deemed suitable for publication in PLOS ONE. Congratulations! Your manuscript is now with our production department. 

Kind regards, 

on behalf of

Professor Federica Canzan 

Academic Editor

PLOS ONE